# DTONet a Lightweight Model for Melanoma Segmentation

**DOI:** 10.3390/bioengineering11040390

**Published:** 2024-04-18

**Authors:** Shengnan Hao, Hongzan Wang, Rui Chen, Qinping Liao, Zhanlin Ji, Tao Lyu, Li Zhao

**Affiliations:** 1Hebei Key Laboratory of Industrial Intelligent Perception, North China University of Science and Technology, Tangshan 063210, China; haoshengnan@ncst.edu.cn (S.H.); m13930201814@163.com (H.W.); zhanlin.ji@gmail.com (Z.J.); 2Changgeng Hospital, Institute for Precision Medicine, Tsinghua University, Beijing 100084, China; cra01052@btch.edu.cn (R.C.); lqpa00594@btch.edu.cn (Q.L.); 3College of Mathematics and Computer Science, Zhejiang A&F University, Hangzhou 311300, China; 4Beijing National Research Center for Information Science and Technology, Institute for Precision Medicine, Tsinghua University, Beijing 100084, China

**Keywords:** lightweight, segmentation, melanoma

## Abstract

With the further development of neural networks, automatic segmentation techniques for melanoma are becoming increasingly mature, especially under the conditions of abundant hardware resources. This allows for the accuracy of segmentation to be improved by increasing the complexity and computational capacity of the model. However, a new problem arises when it comes to actual applications, as there may not be the high-end hardware available, especially in hospitals and among the general public, who may have limited computing resources. In response to this situation, this paper proposes a lightweight deep learning network that can achieve high segmentation accuracy with minimal resource consumption. We introduce a network called DTONet (double-tailed octave network), which was specifically designed for this purpose. Its computational parameter count is only 30,859, which is 1/256th of the mainstream UNet model. Despite its reduced complexity, DTONet demonstrates superior performance in terms of accuracy, with an IOU improvement over other similar models. To validate the generalization capability of this model, we conducted tests on the PH2 dataset, and the results still outperformed existing models. Therefore, the proposed DTONet network exhibits excellent generalization ability and is sufficiently outstanding.

## 1. Introduction

Melanomas, though perceived by some as surface issues, carry significant underlying risks. The skin, being the body’s largest organ, not only plays a protective role in shielding the body from external threats but also serves crucial functions in physiological processes and the immune system. Therefore, any threat to the skin may have profound implications for overall health. The harm caused by skin diseases extends beyond their impact on appearance, affecting the quality of life and mental health of patients [1]. Chronic skin conditions may lead to prolonged pain, itching, and discomfort, significantly impairing daily life and work efficiency. Moreover, the development of some skin diseases may trigger infections and increase the risk of other illnesses. In this context, with the rapid advancement of medical image processing technology, the application of computer-aided diagnostic systems in skin cancer detection has become increasingly important. However, some large image segmentation models have limitations when processing large-scale skin cancer image data, especially in medical scenarios with limited hardware conditions. Because neural networks have advantages over traditional segmentation techniques in dealing with complex problems, we will only discuss neural network methods in this article [2].

Deep learning technology, particularly with convolutional neural networks (CNNs), has achieved remarkable success in skin cancer image segmentation tasks. Models like U-Net [3], Unet++ [4], and Attention U-net [5], while improving in performance, also demand increasing computational resources. The current deep learning models commonly face challenges such as large parameter counts and high computational requirements, restricting their widespread adoption in practical applications. Hence, this study aims to propose a lightweight deep learning model that maintains high accuracy while reducing computational resource demands, thus enhancing its real-time capabilities. By combining deep learning techniques with lightweight strategies, we aim to provide physicians with a convenient and feasible tool to aid in the skin cancer image segmentation process.

In this paper, we will elaborate on the design and training methods of the proposed lightweight model and demonstrate its effectiveness in skin cancer image segmentation tasks through experiments. Through this research, we hope to bring new perspectives to computer-aided diagnostic technology in the medical field, offering a more reliable and efficient solution for early skin cancer diagnosis. Our approach adopts a structure similar to U-Net but introduces several innovations. Firstly, we utilize two different decoders, unlike the traditional U-shaped structure. This enables us to fully leverage the features obtained during the encoding phase, reducing the loss of information in each upsampling step. Secondly, we introduce three modules: the G_O module, G_ECA module, and ORFB module. Integrating these modules into our model allows us to maintain high accuracy with low parameter counts. Therefore, our entire network structure integrates these modules according to our proposed special structure. While maintaining high accuracy, we have reduced our parameter count to an extremely low standard, making our model less hardware-demanding and faster in terms of processing speed.

## 2. Background

With the rapid development of deep learning in recent years, an increasing number of sophisticated models have been widely proposed by researchers. In the field of segmentation, the initial focus was on improving accuracy, but inevitably, this led to an increase in the complexity of models. Therefore, in recent years, researchers have not only been focusing on enhancing accuracy but also optimizing the complexity of models. This initiative has greatly accelerated the process of translating research achievements into practical applications, considering that real-world scenarios may not always have high-performance hardware devices.

Researchers have proposed a variety of segmentation methods for skin lesions to improve the accuracy of the models.

Researchers led by H. Wu [6] proposed a network equipped with an adaptive dual attention module (ADAM), which was proposed to capture the boundary continuity and shape irregularity of skin lesions using global average pooling and pixel-level correlation processing. Q. Zhou [7] and colleagues proposed a superpixel-oriented label distribution learning method, which is used to generate superpixels by a simple linear iterative clustering (SLIC) algorithm, and soft tags are combined with hard tags to enhance segmentation performance. Z. Song [8] and colleagues proposed Res-CDD-Net, which used pre-trained ResNeXt50 to extract rich image features, and they introduced channels and spatial attention blocks to highlight lesion areas and used multi-scale capture blocks to deal with lesions of different sizes. VK Singh [9] and collaborators utilized improved conditional generation adversarial networks (CGans), introduced channel attention decomposition modules (Fcas) to enhance feature differentiation between lesions and non-lesions, and employed multi-scale input strategies to improve the segmentation performance. Y. Liu [10] and colleagues proposed FCP-Net, which combines a mixed loss function and game theory interaction to capture context information and fuse multi-scale features through multiple modules to improve the generalization performance. J. Ruan [11] and collaborators proposed a lightweight model consisting of four modules: DGA (dilated gated attention block), IEA (inverted external attention block), CAB (channel attention bridge block), and SAB (spatial attention bridge block) to improve the competitive skin lesion segmentation performance while reducing parameter and computational complexity. Y. Tang [12] and colleagues proposed MSCGnet, which uses context information to guide feature learning and introduces context-based attention structures into the decoding path to improve the segmentation accuracy. R. Azad [13] and colleagues proposed TransCeption, which utilizes converters and U-shaped network architectures to enhance feature fusion by redesigning modules and introducing multi-scale feature extraction. J. Mu [14] and colleagues proposed M-CSAFN, which uses a multi-color space adaptive fusion network for PWS segmentation, focusing on the internal differences caused by color heterogeneity. D. Dai [15] and colleagues proposed Ms RED, which uses a multi-scale residual coding fusion module and a decoding fusion module to adaptively fuse multi-scale features and introduces multi-resolution and multi-channel feature fusion modules to enhance feature expression. S. Yang [16] and colleagues designed HMT-Net, combined it with Transformer and MLP, and utilized the attention mechanism of the CTrans module to improve the global understanding of lesions. S. Arshad [17] and colleagues proposed Dermo-Seg, which used ResNet-50 and a mixed loss function to effectively detect the differential structures of lesions. H. Yoon [18] and colleagues proposed a deep-learning-based method for the simultaneous segmentation of wrinkles and pores. Unlike color-based skin analysis, this method relies on the analysis of the morphological structure of the skin. The architecture of the model is U-Net, featuring an encoder–decoder structure. Two types of attention mechanisms were added to the network to focus on important regions. Y. Gulzar [19] and colleagues proposed ViT, which focuses on relevant parts of an image by learning long-range spatial relationships. Y. Tang [20] and colleagues proposed a framework using MS-UNet to improve performance through CIFS and deep monitoring mechanisms. J. Wu [21] and colleagues proposed C-UNet, which incorporates Inception convolutional blocks, etc., and uses Dice loss and conditional random fields for optimization. R. Iranpoor [22] and colleagues proposed ResNet101 as the backbone network in a UNet architecture for accurately segmenting lesions from the healthy parts of the skin. R. Wu [23] and colleagues proposed a Transformer-based UNet model for the complex task of segmenting psoriasis lesions from raw color images.

The following is a brief overview of the current research status on lightweight models.

MSF-Net [24] introduces spatial attention mechanisms through residual connections within convolutional blocks, focusing on key regions. Simultaneously, it incorporates multi-scale dilated convolution (MDC) modules and multi-scale feature fusion (MFF) modules to extract contextual information across scales, adaptively adjusting the receptive field size of feature maps. Yuan C [25] proposed UCM-Net, a novel, efficient, and lightweight solution that integrates multilayer perceptrons (MLPs) and convolutional neural networks (CNNs). UCM-Net has fewer than 50 KB parameters and performs at 0.05 gig operations per second (GLOP). Ijaz H [26] introduced a lightweight encoder–decoder deep learning architecture called MobileUNet and EfficientUNet. The encoders are based on the bottleneck blocks of MobileNetV2 and the MBConv blocks of EfficientNetB0, while the decoder follows the structure of the baseline UNet model. Parallel asymmetric convolution (PAC) modules [27] can also be used to replace the traditional square convolution for feature extraction, or LSR [28] can be introduced in the encoder to reduce the number of parameters. Wei S [29] proposed SRP&PASMLP-Net, which focuses on structural reparameterization and parallel axis displacement multilayer perceptrons (MLPS) for robust segmentation performance and fast inference. They introduced reparameterized multiple convolution (RDC) at an early stage to enrich the feature space. Wang Y [30] designed MAUNext, which emphasizes lightweight backbone network design and incorporates skip connections for multi-scale attention mechanisms. The core modules include multi-scale attention convolution, cooperative neighborhood attention MLP coding, and micro-skip join cross-layer semantic fusion for segmentation tasks.

## 3. Materials and Methods

In this chapter, we will give an introduction to our network structure, including the overall network structure, each of our innovative structures, and their combination and principle.

### 3.1. Overall Structure

Our model improves upon the classical encoder–decoder structure, consisting of one encoding stage and two decoding stages. The two decoders, initialized with different weights, decode the results obtained in the decoding stages, resulting in outcomes with different emphases. This significantly enhances the training speed of our model. These outcomes are further fused in the final step to obtain a result closer to the standard answer.

We proposed three modules in total and incorporated them into the new structure, forming our entire network model as shown in the Figure 1.

### 3.2. G_O Block

Images contain both smooth variations and detailed information. Similarly, input and output features in convolutional neural networks (CNNs) include high-frequency and low-frequency components. Low-frequency components contain redundant background information, while high-frequency components retain critical target information. To effectively handle this information, there is no need to store low-frequency features at the same resolution as high-frequency information. This way, we can suppress background interference in low-frequency features, improving the network efficiency. This is the idea behind octave convolution.

In this model, we emulate the structure of octave convolution, replacing the convolutions with ghost convolutions. This ensures a high level of information retrieval and filtering performance while further reducing the required parameter count. The module uses a hyperparameter α to control the ratio of high-frequency to low-frequency components.

The goal of this module is to enhance the convolutional neural network’s (CNN) perception of information at different scales, thus improving the image recognition performance.

In traditional CNNs, convolution operations use the same kernels to process features at all scales. However, in actual images, features at different levels may have different scales. OctaveConv addresses this by introducing two sub-networks, namely, the high-frequency subnetwork and the low-frequency subnetwork, with each responsible for processing high-frequency and low-frequency information, respectively [31].

The main idea and formula of OctaveConv are as follows:

Input: Assume the input feature map is X, where $X_{H}$ represents the high-frequency sub-space, and $X_{L}$ represents the low-frequency sub-space.

Convolution operation: Define two sets of convolution kernels, one for the high-frequency sub-space and one for the low-frequency sub-space. Let the high-frequency convolution kernel be $K_{H}$ and the low-frequency convolution kernel be $K_{L}$. The convolution operation can be expressed as follows:High-frequency convolution:
(1)$$Y_{H}=X_{H}\;\text{×}\;K_{H}\;$$

Low-frequency convolution:


(2)
$$Y_{L}=X_{L}\;\text{×}\;K_{L}$$


Subspace interaction: Downsample the high-frequency convolution result $Y_{H}$ and add it element-wise to the low-frequency convolution result $Y_{L}$ to obtain the final output.

Output feature map:

(3)$$Y=Y_{L}+UpSample(Y_{H})$$where *UpSample* represents the upsampling operation.

This design allows the network to use smaller convolution kernels when processing high-frequency information to capture detailed information while using larger convolution kernels when processing low-frequency information to obtain more global information. By introducing two sub-networks, OctaveConv effectively enhances the network’s perception of multi-scale information, thereby aiding in improving the image recognition performance.

The overall structure is as shown in the Figure 2.

### 3.3. G_ECA Block

We further enhanced module G_O to enable it to serve as an extended encoder, extracting more information. Specifically, we fused module One with efficient channel attention (ECA) [32]. This integration enhances the overall module’s perception of inter-channel correlations, facilitating a more focused utilization of specific channel information in our encoding layer. By replacing the entire module with our encoding layer, we enable different layers to improve inter-channel correlations, thereby enhancing the overall network’s performance and efficiency.

The ECA module is a mechanism designed to enhance a neural network’s attention to channel-wise features. The purpose of this module’s design is to increase the model’s focus on different channel features, allowing for a more effective capture of important information in the image. The introduction of the ECA module primarily aims to achieve advantages in computational efficiency.

The efficient channel attention (ECA) module is a lightweight attention mechanism designed to enhance a neural network’s attention to channel-wise features. Here is a detailed explanation of the ECA module, including some formulaic descriptions.

The input to the ECA (efficient channel attention) module is the feature map X, which contains *C* channels, height *H*, and width *W*. The output is the feature map *Y* with the same dimensions.

Global average pooling: Perform global average pooling on the input feature map, obtaining the average value for each channel. Let the output of the average pooling operation be *Z*, and let the dimensions of *Z* be *C* × 1 × 1.
(4)Z=AvgPool(X)

Here, *AvgPool* represents global average pooling.

Channel attention mapping: Input the output *Z* of the average pooling into a fully connected layer with learned weight parameters, producing channel attention mapping. Let the weight be $W_{attention}$, the bias be $b_{attention}$, and the output of channel attention mapping be: (5)$$A=\sigma (W_{attention}*Z+b_{attention})$$

Here, σ represents the sigmoid function.

Channel-wise weighting: Multiply the channel attention mapping A with the input feature map X, weighting the features on each channel. Obtain the final output feature map Y as follows: (6)Y=A⊙X

Here, ⊙ represents element-wise multiplication.

In summary, the ECA module enhances channel attention in a lightweight manner by introducing global average pooling, channel attention mapping, and channel-wise weighting mechanisms. This module is typically embedded in different layers of deep convolutional neural networks to improve the network’s focus on different channel features, especially in cases where computational resources are limited.

The overall structure is as shown in the Figure 3.

### 3.4. ORFB Block

Our module is also an improvement based on the G_O module, making it more capable of unleashing the maximum performance of our network. Our third module combines the features of the first module and the RFB (receptive field block) module to obtain a larger receptive field [32]. This allows our encoding layer to focus on information contained in a wider range of images and then extract it. However, considering that this module will generate a huge number of parameters, we chose to use only one instance of this module after comprehensive consideration. We placed it in the third layer of the encoding layer to ensure maximum accuracy gain while minimizing the increase in parameter count.

The RFB module is a feature extraction module used for object detection. Its design aims to enlarge the receptive field, enhance the perception of targets, and maintain a relatively low computational cost.

The RFB module is a convolutional neural network (CNN) module designed for object detection. Its purpose is to enhance feature extraction capabilities by introducing non-linear operations and multi-scale information fusion. The following provides a detailed introduction to the RFB module, including some formula explanations.

The input to the RFB (receptive field block) module is the feature map X, which contains C channels, height H, and width W. The output is the feature map Y with the same dimensions.

Local feature map processing: Firstly, perform a 1 × 1 convolution operation on the input feature map, generating a non-linear transformation in the channel dimension. Let the weights be denoted as $W_{non-linear}$ and the bias as $b_{non-linear}$. The output of the non-linear transformation is as follows: (7)$$U=RELU{(W}_{non-linear}*X+b_{non-linear})$$

Here, ∗ represents the convolution operation, and RELU denotes the rectified linear unit.

Multi-scale information fusion: Building upon the non-linear transformation, introduce convolution kernels of multiple scales to perform convolution operations on the feature map. These kernels have different receptive fields, allowing the module to operate on the feature map at different scales. Let the weights of these convolution kernels be as follows: (8)$$W_{scale}^{i}\;\;,i=1,2,\ldots ,N$$

*N* is the number of scales. The output of multi-scale information fusion is as follows:(9)$$V=\sum_{i=1}^{n}W_{scale}^{i}*U$$

Here, ∗ denotes the convolution operation, and V is the fused feature map.

Channel attention mapping: For the fused feature map V, perform global average pooling to obtain the average value for each channel. Let the weight of the channel attention mapping be $W_{attention}$. The output of the channel attention mapping is as follows: (10)$$A=\sigma \left(W_{attention}*\left(AvgPool\left(V\right)\right)\right)$$

Here, σ represents the sigmoid function. 

Channel-wise weighting: Finally, multiply the channel attention mapping A with the fused feature map V, weighting the features on each channel. Obtain the final output feature map Y as follows: (11)Y=A⊙V

Here, ⊙ represents element-wise multiplication.

In summary, the RFB module enhances the perception of targets by introducing non-linear transformation, multi-scale information fusion, and channel attention mechanisms. This module is typically embedded in deep neural networks to extract more representative features, especially achieving good performance in object detection tasks.

The overall structure is as shown in the Figure 4.

## 4. Experiments and Results

### 4.1. Dataset

In this experiment, we utilized two datasets to validate the effectiveness of our model. The first dataset was ISIC2018, which is a large-scale dataset for dermatopathology research provided by the International Skin Imaging Collaboration (ISIC). We referred to the description and processing method of this dataset by Ali et al. [33], and finally decided to use one of the data enhancement methods of random HueSaturationValue, RandomBrightness, and RandomContrast for our dataset. This dataset comprises skin images from various skin diseases and conditions, including malignant melanoma (melanoma) and benign lesions. We specifically focused on the labeled images of melanoma, dividing them into training, testing, and validation sets in a ratio of 7:1:2.

The second dataset used was PH2, a dermatopathology image dataset developed by the University of Porto in Portugal. This dataset is dedicated to the study of malignant melanoma and benign lesions. PH2 includes macroscopic images from over two-hundred patients, hundreds of digital images, and corresponding pathological images. It provides high-resolution images for in-depth research into the structure and features of skin lesions.

### 4.2. Evaluation Metrics

In this experiment, we used several metrics to compare with other models, including the following: 

IoU (intersection over union) is a metric widely used in segmentation tasks. In semantic segmentation or object detection, it measures the overlap between the region predicted by the model and the actual region. The IOU value ranges from 0 to 1, where 1 indicates complete overlap and 0 indicates no overlap.
(12)$$IoU=\frac{TP}{TP+FP+FN}$$

The Dice coefficient is another evaluation metric used for segmentation tasks. Similar to IOU, it measures the similarity between the region predicted by the model and the actual region. The Dice coefficient also ranges from 0 to 1, where 1 indicates a perfect match.
(13)$$Dice=\frac{2TP}{2TP+FP+FN}$$

Accuracy (ACC) measures the proportion of correctly predicted pixels among all pixels.
(14)$$Acc=\frac{TP+TN}{TP+FP+FN+TN}$$

Parameter count refers to the number of weights and biases that a model needs to learn. It is a measure of the size of the model. More parameters usually indicate a more powerful model, but they also require more storage and computational resources. Having too many parameters can lead to overfitting.

Floating-point operations (FLOPs) are the total number of floating-point operations executed during the forward pass of a model. FLOPs serve as a metric for measuring the computational complexity of a model. They are crucial for understanding the computational resources required for inference. Lower FLOPs generally indicate a lighter model, suitable for inference in resource-constrained environments.

### 4.3. Experimental Setup

In this study, Python version 3.9 was used, and the Windows 10 operating system was used on a computer equipped with a 12th-generation Intel^®^ Core™ i5-13490 CPU and an NVIDIA GeForce RTX 3060 (12 GB memory) (From Pinduoduo, purchased in Tangshan, China).

We set the initial learning rate to 1 × 10^−4^, with a minimum learning rate of 1 × 10^−5^. The batch size was set to 4, and the number of epochs was 500. The input image dimensions were 256 × 256, and we used the BCEDiceLoss loss function.

### 4.4. Results and Analysis

#### 4.4.1. Melanoma Segmentation Comparison with Advanced Models with ISIC2018

To evaluate the performance of our model, we compared it with several mainstream models in the market. We adopted a method of testing our model by randomly inputting images ten times and taking the average to ensure the absence of random occurrences. Subsequently, we obtained Table 1 and Table 2. Table 1 presents the results of testing on the ISIC2018 dataset, while Table 2 demonstrates the validation on the PH2 dataset to showcase the generalization performance of our model.

As shown in the table above, it can be observed that the parameter count of our model was significantly lower than all the other models, standing at only 0.03. Although the floating-point operation count was not the lowest, it still ranked among the top. Moving on to performance metrics, our model achieved the highest IOU, leading the second-best by 0.0067. The accuracy was also significantly ahead, reaching a maximum of 0.9607. Additionally, the Dice coefficient was at a medium-to-high level.

Therefore, from the data shown in this table, we can conclude that our model not only ranked first among all the models in terms of the accuracy of segmentation but also had the lowest number of parameters, which can further reduce the hardware requirements of our model and make it more widely applicable.

The Figure 5 demonstrate a comparative view between our model’s predicted results and the actual images. For ease of observation, we overlayed them together.

As shown in this Figure 5, we can see that the segmented results of our model had a very high coincidence with the real lesions, which indicates be very accurate segmentation of the diseased area.

#### 4.4.2. Melanoma Segmentation Comparison with Advanced Models with PH2

Looking at the performance of our model on the PH2 dataset, it is evident that our IOU and accuracy remained the highest, and the Dice coefficient was also at a leading level. This further validates the generalization performance of our model, demonstrating its ability to achieve excellent results on different datasets.

The Figure 6 shows the performance of our model on the PH2 dataset, showing that the performance was still stable and excellent.

As shown in the figure above, through our coincidence technique, it is obvious that our segmentation results overlapped perfectly with the lesion area.

In order to better show the segmentation effect of each network, we deliberately captured some of the results and formed a graph with real pictures and the gold standard in Figure 7. From this figure, we can see that each of our models could segment the rough shape, which is enough to prove the excellent performance of the neural network, but in terms of details, it is obvious that our model had better processing performance and was closer to the gold standard.

#### 4.4.3. Cross-Domain Generalization Validation—Breast Ultrasound Dataset

To test the performance of our model in terms of cross-domain generalization, we tested it on a breast ultrasound dataset. The results were compared with those of some other models. The results are shown in Table 3.

From the data in this table, we can see that the accuracy of our model for breast cancer segmentation is only comparable to that of the classical model in terms of cross-domain generalization. Even though our model is small in terms of the number of parameters, we still have a big gap compared with other advanced models in this field. Therefore, in terms of cross-domain generalization, our model still has huge room for improvement. The future improvement goal is to create a module that can capture different features while keeping the number of parameters low to deal with this problem.

## 5. Discussion

Through our research, we successfully proposed and developed a lightweight model focused on the task of melanoma segmentation. The model maintains high accuracy while demonstrating excellent computational efficiency, making it suitable for real-time segmentation in scenarios with limited computing resources. By cleverly introducing lightweight modules, we effectively reduced the model’s parameter count and computational burden, making it a viable solution widely applicable in embedded devices and mobile applications.

In various dataset evaluations, our lightweight model showed outstanding performance and demonstrated strong generalization capabilities across different data distributions. This indicates that our model has strong adaptability and can robustly perform melanoma segmentation tasks in diverse clinical scenarios. This achievement provides significant support for advancing the early detection of melanoma in the healthcare sector, with the potential to offer patients more timely and effective treatment.

Future research directions will focus on further improving the lightweight performance of the model to meet the requirements of embedded devices and mobile applications. We plan to achieve an ideal balance between performance and efficiency by delving into the balance of network structure, parameter count, and computational complexity. This will provide valuable experience and guidance for the development of lightweight models in medical image segmentation tasks.

## Figures and Tables

**Figure 1 bioengineering-11-00390-f001:**
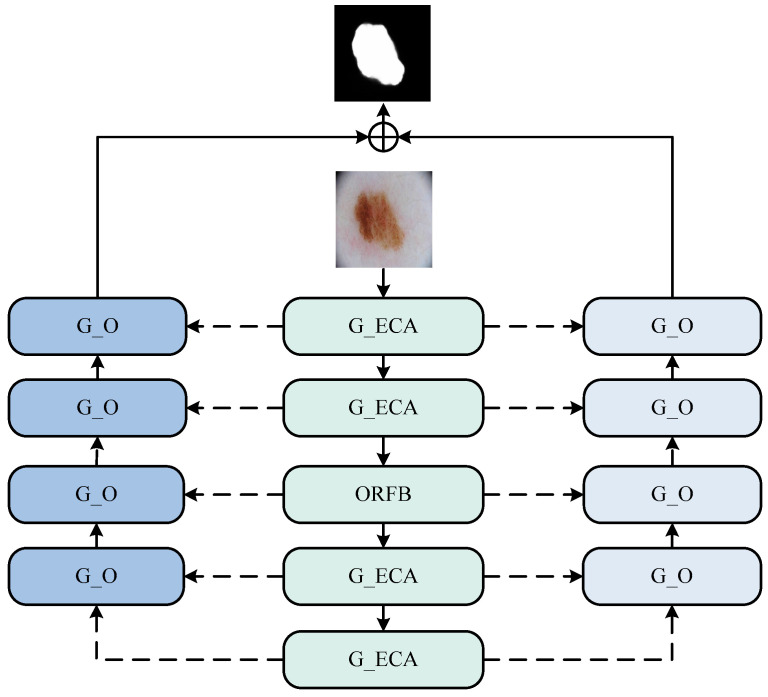
The DTO-Net network. (This network consists of an encoding phase and two decoding phases. The encoding phase comprises G_O (ghost octave) and ORFB (octave receptive field block) modules, while the decoding phase consists of the G_ECA (ghost efficient channel attention) module).

**Figure 2 bioengineering-11-00390-f002:**
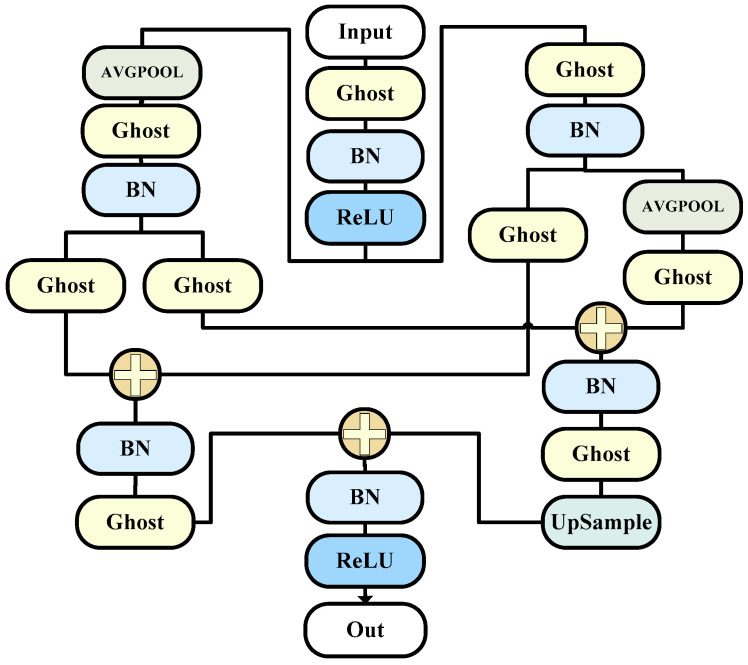
The G_O model (this is the specific composition diagram of the corresponding decoder).

**Figure 3 bioengineering-11-00390-f003:**
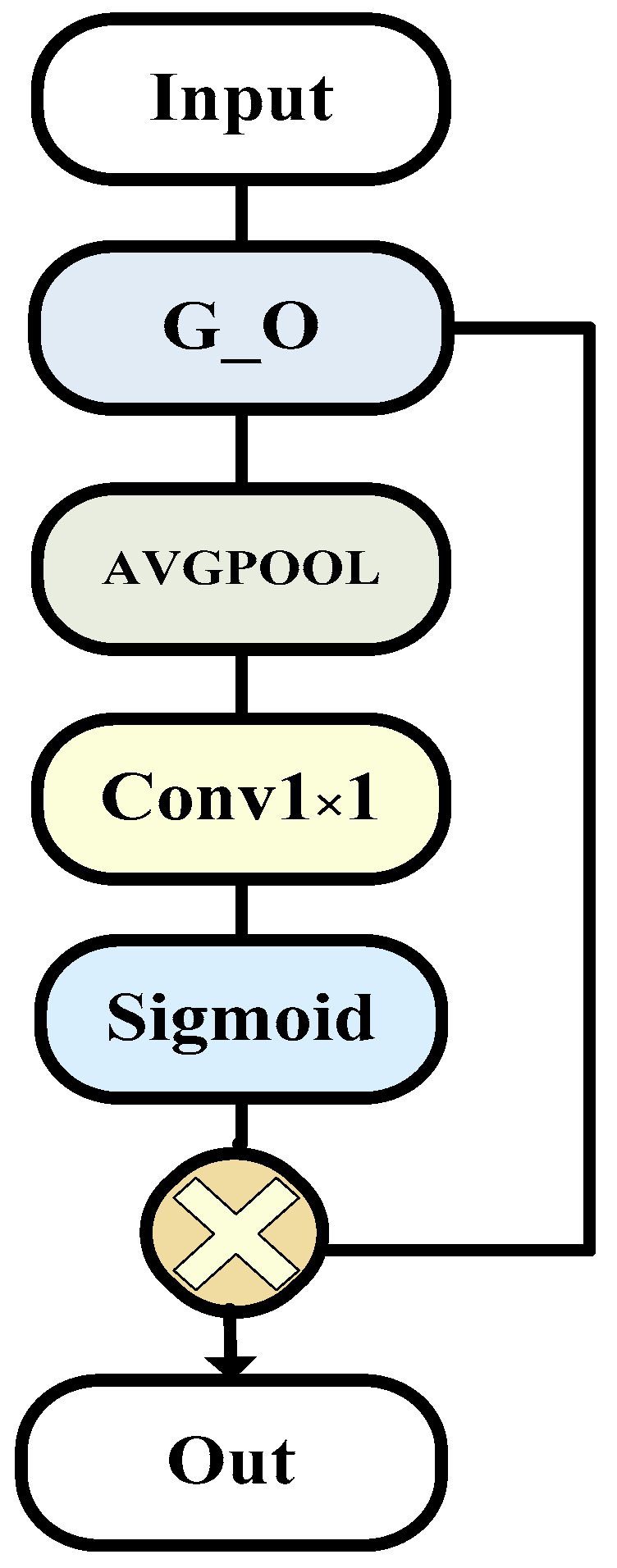
The G_ECA model (this is the specific composition of encoder component module G_ECA, which is improved on the basis of G_O module to make it more suitable for encoders).

**Figure 4 bioengineering-11-00390-f004:**
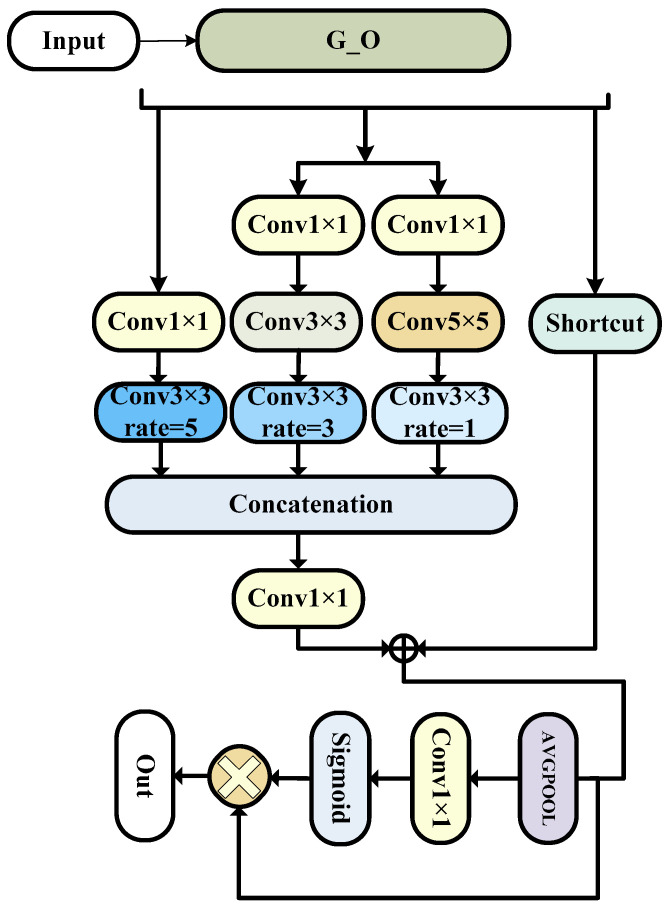
The ORFB model (this is the specific composition of the encoder component module ORFB, which is improved on the basis of the G_O module, which can improve the accuracy very well, but it will produce too many parameters, so it is only used here).

**Figure 5 bioengineering-11-00390-f005:**
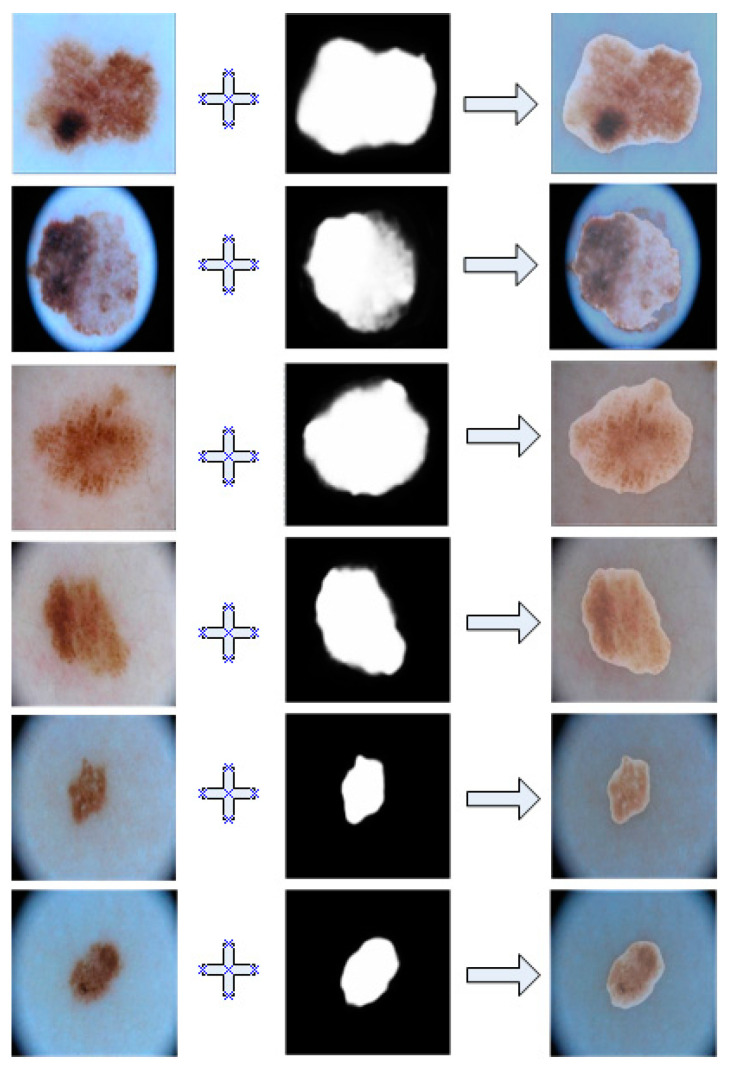
Overlayed image of segmentation results and ground truth in ISIC2018.

**Figure 6 bioengineering-11-00390-f006:**
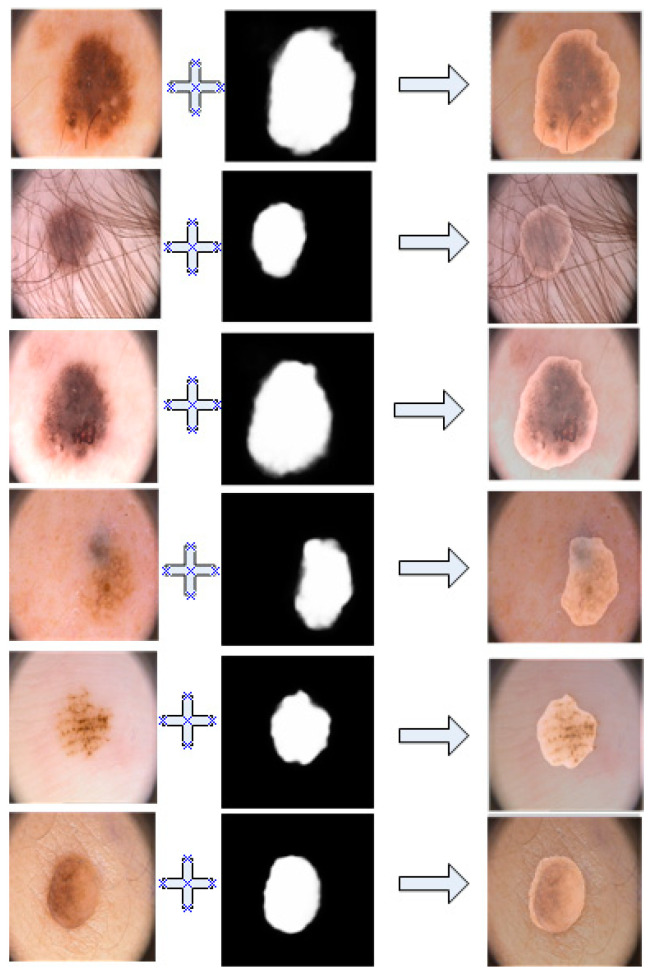
Overlay image of segmentation results and ground truth in PH2.

**Figure 7 bioengineering-11-00390-f007:**
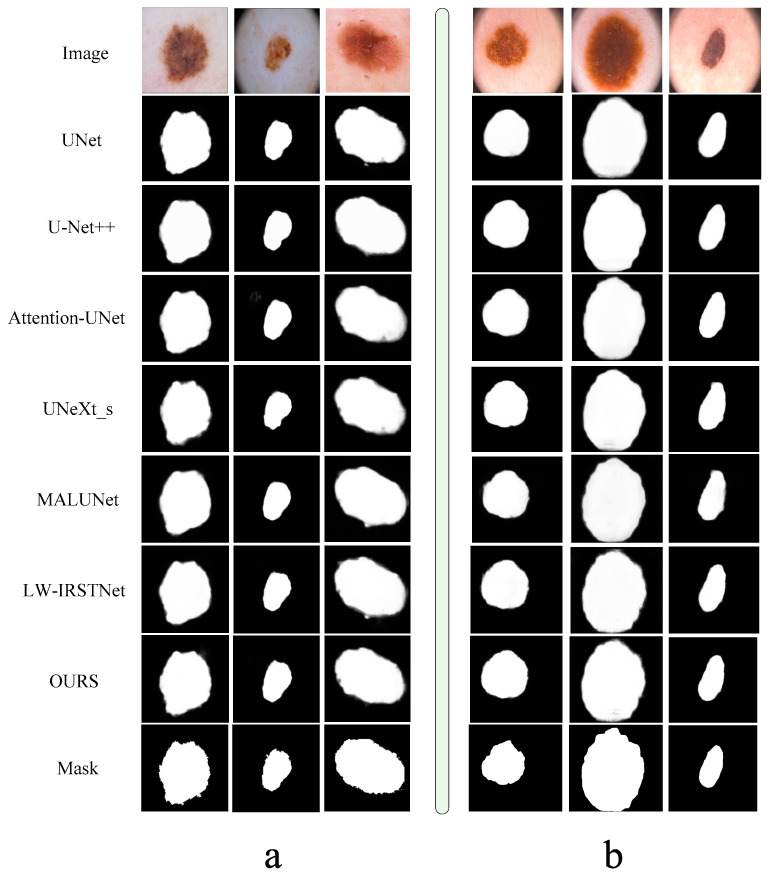
Comparison of segmentation results of each model. The left part (**a**) is the presentation of results in the ISIC2018 dataset, and the right part (**b**) is the presentation of segmentation results in the PH2 dataset.

**Table 1 bioengineering-11-00390-t001:** Segmentation results of advanced models.

Models	Parameters	GFLOPS	IoU	Acc	Dice
UNet	7.770	13.780	0.8169	0.9576	0.8838
U-Net++	9.160	34.900	0.8203	0.9587	0.8881
Attention-UNet	8.730	16.740	0.8217	0.9588	0.8863
UNeXt_s [34]	0.300	0.100	0.8057	0.9557	0.8895
MALUNet [11]	0.175	0.083	0.8120	0.9532	0.8924
LW-IRSTNet [35]	0.161	0.301	0.8216	0.9588	0.8854
Ours	0.030	0.126	0.8284	0.9607	0.8845

**Table 2 bioengineering-11-00390-t002:** Segmentation results of advanced models in PH2 dataset.

Models	Parameters	GFLOPS	IoU	Acc	Dice
UNet	7.770	13.780	0.8062	0.9276	0.8916
U-Net++	9.160	34.900	0.7929	0.9238	0.8831
Attention-UNet	8.730	16.740	0.8102	0.9303	0.8716
UNeXt_s	0.300	0.100	0.8077	0.9277	0.8900
MALUNet	0.175	0.083	0.8278	0.9351	0.9048
LW-IRSTNet	0.161	0.301	0.8327	0.9386	0.8944
Ours	0.030	0.126	0.8347	0.9388	0.8914

**Table 3 bioengineering-11-00390-t003:** Segmentation results in breast ultrasound dataset.

Models	IoU	Acc	Dice
UNet	0.6859	0.9642	0.7913
U-Net++	0.6861	0.9625	0.7900
Attention-UNet	0.6875	0.9636	0.7948
Ours	0.6901	0.9633	0.7910

## Data Availability

Data are contained within the article.

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
