# Peer review of "DTONet a Lightweight Model for Melanoma Segmentation"

_bioengineering, 2024, doi:10.3390/bioengineering11040390_

Round 1

Reviewer 1 Report

Comments and Suggestions for Authors

SUMMAR

The authors proposed a new skin disease segmentation model based on the encoder-decoder structure and n convolutional layers. The paper is well-structured. It provides method description and experiment results. The comparison with other models using a benchmark dataset is presented.

COMMENTS

1. Please, show more examples of correct and incorrect segmentation.

2. Why did you refuse to use an augmentation? 

3. More detailed analysis should be provided to explain why the proposed model showed the best results. 

4. Please preview the results of other authors on the datasets used or in similar tasks. This is necessary to understand the state-of-the-art accuracy in this problem.

5. How can the model be used if a melanoma is not localized (does not occupy most of the image area and is not located in the center)?

Reviewer 2 Report

Comments and Suggestions for Authors

A general comment is that the paper refers only to neural network segmentation but does not review or discuss conventional medical segmentation methods, such as contrast,  colour or edge-based methods. Why is it so? Is it because neural methods perform better? If so, some literature review should be provided to justify this point.

In Section 2, several acronyms are undefined. E.g. IEA, DGA, CAB, SAB, and others.

Section 2 is overly convoluted and requires better organisation. First, it contains extremely long paragraphs. Second, some classification of methods would help the reader understand the taxonomy of the reviewed methods.

Section 3 deserves an introduction. I would recommend that this introduction provides an overview that guides the reader to what follows next.

In line 223, “Nature images” is probably a wrong term. 

Please provide some intuition as to what a “G_O block” is.

The block names in Fig. 1 are undefined.

The block names in Fig. 2 are undefined.

Equations (1) and (2) appear to be the same. Why don't you merge them into one? (I understand that they are two separate operations but the only thing that changes is the kernel size).

ECA is defined twice in line 276 and 281 with different definitions.

Isn't the symbol defined in 302 the same as convolution? If so, is it needed?

AvgPool was defined too late after Equation 10, while already used in Equation 4.

Fig 4, is poorly drafted and arrows should be used throughout the data path.

The benchmark datasets are not cited.

Besides the GFLOPs please report some indicative execution times on a regular computer as the ones you mention in the abstract that the hospitals have. This way the reader can better appreciate the value of the proposed contribution.

Minor:

A space character is needed after each period (‘.’).

A space character is needed before each citation bracket (‘[‘).

Line 108. Delete one period from “..”.

Reviewer 3 Report

Comments and Suggestions for Authors

This manuscript introduced a lightweight deep-learning algorithm for skin lesion segmentation. The proposed algorithm aims to maintain high segmentation accuracy and require less computation time.

Major issue: I don’t see any real clinical implication for this approach, especially for melanoma lesions. The melanoma lesion has very good contrast in the photos. A simple thresholding method could easily segment the lesion. What’s the purpose of designing a complicated deep learning approach for this? 

The author doesn't seem to have any background on melanoma. How important is it to segment the surface shape of melanoma? Because melanoma can penetrate and spread deeply into the skin, which makes it the most lethal skin cancer. 

Minor issue: 

1. Although the title says skin disease segmentation, the author only tested the network on melanoma. The title is highly misleading!

2. What exactly is figure 1? It needs a better description.

3. All the figure titles need more information.

4. Section 2 background, line 78 to 170, line 178 to 210, these two paragraphs are sooooo long! It’s very difficult to read and follow. Please re-organize them.

5. What do “G_O” and ”G_O_ECA” stand for, in figure1?

6. Title 3.3 G_ECA? Is this correct? In figure 1, it’s G_O_ECA

Round 2

Reviewer 2 Report

Comments and Suggestions for Authors

I find that the issues found in the review have been resolved and that the paper merits publication.

Author Response

Thank you for your affirmation!

Reviewer 3 Report

Comments and Suggestions for Authors

No significant improvement for the manuscript. The whole experiment design has flaw. This is just a meaningless approach for melanoma segmentation, that really has no clinical value no matter what the author claims. 
